# ID-COMPOSER: MULTI-SUBJECT VIDEO SYNTHESIS WITH HIERARCHICAL IDENTITY PRESERVATION

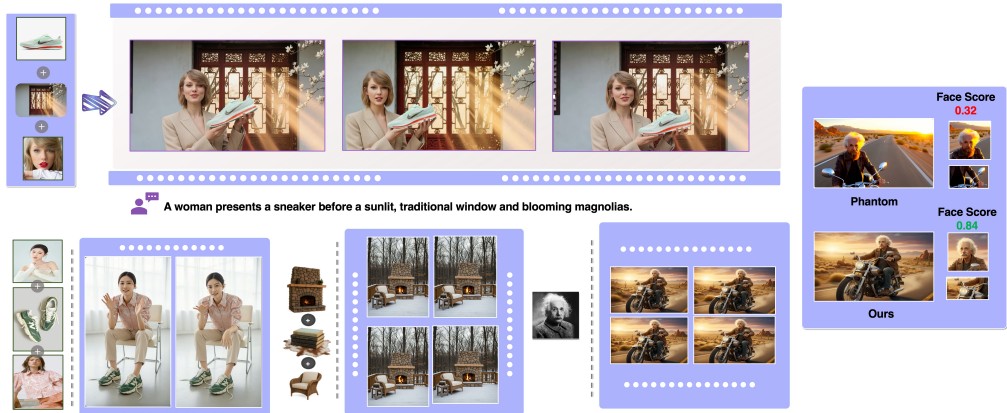

Figure 1: Given a text prompt and multiple reference images, ID-COMPOSER generates subject-consistent videos and achieves impressive subject ID preservation compared with the previous state-of-the-art methods, such as Phantom (Liu et al., 2025b). Kindly zoom in for details.

## ABSTRACT

Video generative models pretrained on large-scale datasets can produce high-quality videos, but are often conditioned on text or a single image, limiting controllability and applicability. We introduce ID-COMPOSER, a novel framework that addresses this gap by tackling multi-subject video generation from a text prompt and reference images. This task is challenging as it requires preserving subject identities, integrating semantics across subjects and modalities, and maintaining temporal consistency. To faithfully preserve the subject consistency and textual information in synthesized videos, ID-COMPOSER designs a **hierarchical identity-preserving attention mechanism**, which effectively aggregates features within and across subjects and modalities. To effectively allow for the semantic following of user intention, we introduce **semantic understanding via pretrained vision-language model (VLM)**, leveraging VLM's superior semantic understanding to provide fine-grained guidance and capture complex interactions between multiple subjects. Considering that standard diffusion loss often fails in aligning the critical concepts like subject ID, we employ an **online reinforcement learning phase** to drive the overall training objective of ID-COMPOSER into RLVR. Extensive experiments demonstrate that our model surpasses existing methods in identity preservation, temporal consistency, and video quality. Code and training data will be released.

## 1 INTRODUCTION

Recent advances in video generative models (OpenAI, 2023; Yang et al., 2024b; Vidu, 2024; keling, 2024; Wan, 2025) achieve high-fidelity results, yet their reliance on sparse inputs, typically a text prompt or a single initial frame, severely restricts controllability. Inspired by the compositional flexibility of modern image generators (Deng et al., 2025a; Fortin et al., 2025), we address this

limitation by focusing on multi-subject video generation. This capability further facilitates downstream applications in subject-driven video synthesis, dynamic scene composition, and controllable product placement, and promises to advance personalized content creation, virtual storytelling, and advertising. Despite its impact, multi-subject video generation still faces challenging issues compared to the existing pipeline established for generating single-subject video: (1) **Identity preservation** is increasingly difficult for multiple subjects throughout the video; (2) **Semantic integration** on more subjects and modalities involves challengingly aligning and balancing semantic information; (3) **Temporal consistency** of multi-subject video demands preserving coherent appearance and motion across consecutive frames. To address it, a naive solution is to develop an agentic system that first leverages off-the-shelf subject-driven image generation models (Ruiz et al., 2023) to generate the first frame given the text prompt and reference images, and then applies video generative models (OpenAI, 2023; Yang et al., 2024b; Wan, 2025) to produce the subsequent frames. However, this two-stage pipeline lacks explicit correspondence between the video and the reference subjects, and heavily depends on the quality of the initial frame, resulting in suboptimal identity preservation, poor temporal consistency, and vulnerability to error accumulation. To tackle these challenges, recent works (Deng et al., 2025b; Liu et al., 2025b; Xue et al., 2025a) propose end-to-end solutions by injecting information of multiple subjects into the Multimodal Diffusion Transformers (MMDiT) (Esser et al., 2024) of pretrained video diffusions (Wan, 2025). Although effective, these methods simply concatenate visual tokens of all subjects and textual tokens to noise tokens. This results in higher perplexity during semantic integration and conflict resolution across different subjects, as well as suboptimal performance due to lacking explicit modeling of interactions among subjects and modalities.

To this end, we propose ID-COMPOSER, a novel framework for multi-subject video generation that achieves state-of-the-art performance in identity preservation, temporal consistency, and overall video quality. The key designs of ID-COMPOSER are twofold: (1) **A hierarchical identity-preserving attention mechanism.** The task we are addressing involves multiple subjects and modalities, making it crucial to effectively aggregate features both within and across different conditions. Inspired by recent work using local-global attention (Wang et al., 2025a; Lin et al., 2025), we design a hierarchical attention mechanism that performs attention on three progressive levels: intra-subject, inter-subject, and cross-modal attention. This design enables effective identity consistency and textual faithfulness by capturing fine-grained details within each subject, modeling interactions among multiple subjects, and integrating information across modalities. (2) **Semantic understanding via pretrained vision-language model (VLM).** Our model leverages VLM as a text encoder to provide fine-grained guidance and capture complex interactions between multiple subjects. Compared to traditional text encoders (Radford et al., 2021), VLMs (Yang et al., 2024a) demonstrate superior semantic understanding and cross-modal alignment capabilities, which are crucial for the multi-subject video generation. ID-COMPOSER is built upon a strong video diffusion model (Wan, 2025) and incorporates the above two designs into the MMDiT (Esser et al., 2024) architecture, enabling effective generation of high-quality videos with multiple specific subjects. To incentivize the model to generate videos with high quality and strong identity preservation, we further introduce **an online reinforcement learning** (Liu et al., 2025a; Guo et al., 2025) phase subsequent to the initial diffusion training. A new dataset is constructed to facilitate the training and evaluation of multi-subject video generation models. We introduce a data-curation pipeline leveraging vision-language models and advanced image editing models. Our process involves automated filtering, captioning, and subject-decomposing stages to process video clips, resulting in a mixed-modality dataset.

As shown in Fig. 1, ID-COMPOSER is capable of generating high-quality videos that accurately preserve the identities of multiple subjects, align with the prompt, and maintain temporal consistency. Experiments demonstrate that our model surpasses existing methods in identity preservation, temporal consistency, and video quality. Ablation studies validate the effectiveness of our proposed designs.

In summary, our contributions are as follows:

- We propose ID-COMPOSER, a novel framework for multi-subject video generation that effectively preserves identities, integrates semantics, and maintains temporal consistency.

- Our proposed hierarchical identity-preserving attention mechanism, the use of pretrained VLM for semantic understanding, and the incorporation of online reinforcement learning significantly enhance the model's performance in handling multiple subjects and modalities.

- We construct a new dataset for multi-subject video generation, and conduct extensive experiments demonstrating the superiority of ID-COMPOSER over existing methods.

## 2 RELATED WORKS

**Subject-Driven Image Generation**   Recent advances in subject-driven image generation have enabled the creation of high-fidelity visual assets, profoundly impacting digital content creation. A prominent line of work involves tuning-based methods, which achieve high subject fidelity by fine-tuning a generative model on a small set of reference images. These approaches range from optimizing the entire model's weights, as in DreamBooth (Ruiz et al., 2023), to learning subject-specific textual embeddings (Gal et al., 2022) or adopting parameter-efficient fine-tuning strategies (Hu et al., 2021; Han et al., 2023; Yuan et al., 2023). However, a significant drawback of these methods is the substantial computational overhead required for per-subject optimization. To circumvent this limitation, tuning-free methods (Ye et al., 2023; Wang et al., 2024; Sun et al., 2024; Wu et al., 2024; Liu et al., 2025c) have been proposed, which inject identity information during inference via various conditioning mechanisms, thereby offering a more efficient alternative. More recently, a new paradigm has emerged with versatile, unified models (Chen et al., 2024; Xiao et al., 2024; Lu, 2024; Wu et al., 2025a; Gao et al., 2025; Fortin et al., 2025) that incorporate subject-driven generation as one of many capabilities. Despite this progress, extending subject-driven generation from the image to the video domain remains a formidable challenge, primarily due to the difficulty of maintaining both identity and temporal consistency across frames.

**Subject-Consistent Video Generation**   Subject-consistent video generation (keling, 2024; Vidu, 2024; Pika, 2024; Jiang et al., 2025) is commonly achieved by enhancing the model's attention mechanisms to incorporate appearance cues. More flexible paradigms have also emerged, including adapter-based methods like ID-Animator (He et al., 2024), ConsisID (Yuan et al., 2024), and Stand-In (Xue et al., 2025a). For the specific challenge of handling multiple subjects, a router mechanism has been introduced (Mou et al., 2025). Other works have explored richer customization dimensions: Wang et al. (2025b) design FantasyPortrait to enhance multi-character portrait animation through expression-augmented diffusion transformers. Hu et al. (2025) propose PolyVivid to improve vividness and consistency in multi-subject scenarios via cross-modal interaction. Huang et al. (2025) present Videomage, which enables both multi-subject and motion customization in text-to-video diffusion. Our work is most closely related to Phantom (Liu et al., 2025b), Concat-ID (Zhong et al., 2025), SkyReels-A2 (Fei et al., 2025), and CINEMA (Deng et al., 2025b), which focus on attention-based feature injection. However, these methods, as well as the above extensions, often struggle with conflicts between subject identity and textual prompts and with maintaining consistency. Moreover, ID-COMPOSER proposes a novel method to mitigate these conflicts and enforce multi-level consistency, thereby improving the robustness and quality of multi-subject video generation.

**Reinforcement Learning for Generation**   Leveraging its computational efficiency, Direct Preference Optimization (DPO) (Gao et al., 2025) has become a prevalent technique for post-training text-to-image models to align them with human preferences, such as aesthetics and prompt fidelity. Subsequent work, like DenseDPO (Wu et al., 2025b), extends this paradigm to the video domain by refining paired data construction and the granularity of preference labels, yielding videos with superior visual quality and motion dynamics. A primary limitation of these methods, however, is their offline nature, which precludes online parameter updates. This challenge has been recently addressed by verifiable reward-based models like OpenAI's o1 (Jaech et al., 2024) and Deepseek-R1 (Guo et al., 2025). Concurrently, the growing interest in online learning has spurred various online Reinforcement Learning (RL) methods, including ReFL (Xu et al., 2023), DRaFT (Clark et al., 2023), FlowGRPO (Liu et al., 2025a), DanceGRPO (Xue et al., 2025b), and SRPO (Shen et al., 2025), which have demonstrated promising results in enhancing image aesthetics and text-rendering fidelity. Nevertheless, their application to subject-consistent video generation remains limited. To bridge this gap, we introduce task-specific rewards tailored for subject-consistent video generation, a method that mitigates reward hacking and delivers videos with high identity consistency.

## 3 METHOD

We are addressing the task of multi-subject video generation. Given a textual prompt $\mathbf{C}_{\text{txt}}$ and a set of $N$ reference images $\mathcal{I} := \{\mathbf{I}_k\}_{k=1}^{N}$, where each $\mathbf{I}_k$ depicts a unique subject, the objective is to generate a video $\mathbf{V}$ that aligns with the prompt $\mathbf{C}_{\text{txt}}$ and faithfully preserves the identities of all $N$ subjects from $\mathcal{I}$ with high fidelity and temporal coherence. In the following sections, we first delineate the preliminaries of our method (Sec. 3.1) and subsequently introduce the architecture of the proposed method (Sec. 3.2), training schemes (Sec. 3.3), and the data curation pipeline (Sec. 3.4).

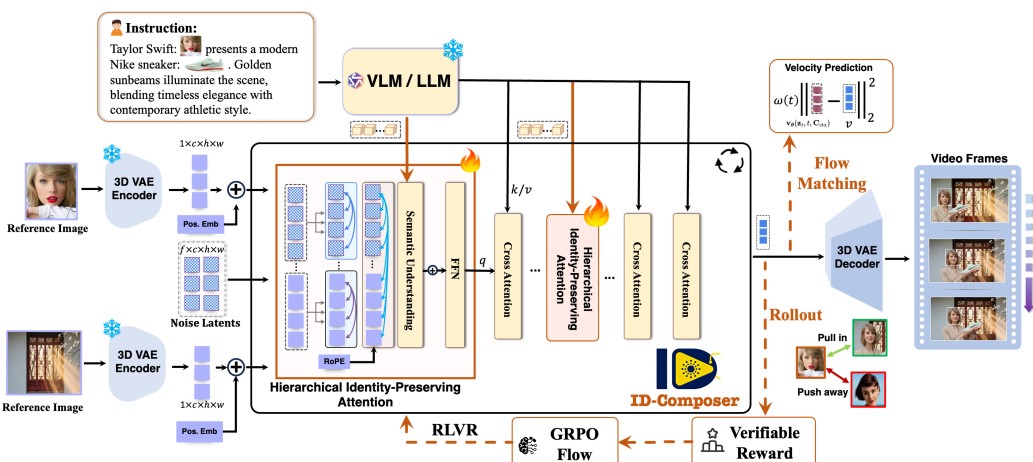

Figure 2: **ID-COMPOSER Overview**. Our model incorporates a hierarchical identity-preserving attention mechanism and a VLM that performs reasoning on the multimodal input into a video DiT to enable multi-subject video generation. An online RL stage further refines the concept alignment.

## 3.1 PRELIMINARIES: RECTIFIED FLOW FOR VIDEO GENERATION

Our methodology is founded upon a latent video diffusion transformer (Peebles & Xie, 2023; Wan, 2025) trained with Rectified Flow (RF) (Lipman et al., 2022; Liu et al., 2022; Esser et al., 2024). In latent video diffusion, a video clip $\mathbf{V} \in \mathbb{R}^{T \times H \times W \times 3}$ is first projected into a latent space by the encoder $\mathcal{E}$ of a variational autoencoder (VAE), yielding the representation $\mathbf{z}_0 := \mathcal{E}(\mathbf{V}) \in \mathbb{R}^{f \times c \times h \times w}$. The corresponding VAE decoder is denoted by $\mathcal{D}(\mathbf{z}_0)$. The generative process is formulated based on Rectified Flow, which defines a straight-line trajectory between a sample from the data distribution $\mathbf{z}_0 \sim p_{\text{data}}$ and a sample from a simple prior distribution, typically a standard Gaussian $\boldsymbol{\epsilon} \sim \mathcal{N}(\mathbf{0}, \mathbf{I})$. The trajectory is parameterized by a time variable $t \in [0, 1]$:

$$\mathbf{z}_t := (1 - t)\mathbf{z}_0 + t\boldsymbol{\epsilon}. \tag{1}$$

This path induces a constant velocity field $\mathbf{v} = \boldsymbol{\epsilon} - \mathbf{z}_0$. The core objective is to train a video diffusion transformer $\mathbf{v}_{\boldsymbol{\theta}}$, parameterized by $\boldsymbol{\theta}$, to predict this velocity. The training objective is formulated as a regression loss over this velocity field:

$$\mathcal{L}_{\text{RF}} := \mathbb{E}_{t, \mathbf{z}_0, \boldsymbol{\epsilon}} \left[ w(t) \left\| \mathbf{v}_{\boldsymbol{\theta}}(\mathbf{z}_t, t, \mathbf{C}_{\text{ctx}}) - (\boldsymbol{\epsilon} - \mathbf{z}_0) \right\|_2^2 \right], \tag{2}$$

where $\mathbf{C}_{\text{ctx}}$ represents the conditioning information (e.g., a text or image prompt), and $w(t)$ is a weighting function that balances the loss across different noise levels.

## 3.2 ID-COMPOSER: MULTI-SUBJECT VIDEO GENERATION

As shown in Fig. 2, ID-COMPOSER extends the DiT-based latent video diffusion architecture by introducing two key architectural innovations to handle the complexities of multi-subject conditioning: (1) **A hierarchical identity-preserving attention mechanism**, which is designed to hierarchically aggregate features. (2) **Semantic understanding via a pretrained Vision-Language Model (VLM)**, which we employ as a sophisticated encoder for both textual and visual inputs. To further incentivize the model's semantic alignment and identity preservation, we incorporate **an online reinforcement learning phase** subsequent to the initial flow matching training.

### 3.2.1 MODEL ARCHITECTURE

**Hierarchical Identity-Preserving Attention.** Our model is based on multi-modal DiT (MMDiT), where latent video tokens and conditioning tokens are processed together through a series of transformer blocks. Following this, we first encode the input reference images $\mathcal{I}$ using a pretrained image encoder (e.g., Wan-VAE) to obtain visual feature maps $\{\mathbf{F}_k\}_{k=1}^N$, where each $\mathbf{F}_k \in \mathbb{R}^{c \times h \times w}$

corresponds to the $k$-th subject. These feature maps are then flattened into token sequences $\{\mathbf{f}_k\}_{k=1}^N$, with each $\mathbf{f}_k \in \mathbb{R}^{hw \times c}$. The textual prompt $\mathbf{C}_{\text{txt}}$ is encoded using a pretrained text encoder to produce a sequence of text tokens $\mathbf{f}_{\text{txt}} \in \mathbb{R}^{l \times c}$, where $l$ is the number of text tokens. The complete conditioning token set is thus $\mathbf{C}_{\text{ctx}} = [\mathbf{f}_{\text{txt}}; \mathbf{f}_1; \ldots; \mathbf{f}_N]$. To effectively integrate information from multiple subjects and modalities, we introduce a hierarchical attention mechanism within each transformer block. This mechanism operates in three stages: (1) **Intra-subject attention**: Each subject's tokens $\mathbf{f}_k$ undergo self-attention independently, allowing the model to capture fine-grained details and spatial relationships within each subject. (2) **Inter-subject attention**: The outputs from the intra-subject attention are concatenated and subjected to another self-attention layer, enabling the model to learn interactions and relationships between different subjects. (3) **Multi-modal attention**: Finally, the combined subject features are concatenated with the text tokens $\mathbf{f}_{\text{txt}}$ and the latent video tokens, followed by a self-attention operation that fuses information across modalities. This hierarchical approach ensures that the model can effectively preserve individual subject identities while also capturing complex inter-subject dynamics and aligning them with the textual prompt.

**Semantic Reasoning via Vision-Language Models.** While the hierarchical attention mechanism is designed to preserve identity, composing multiple subjects coherently in accordance with ambiguous user instructions remains a formidable challenge. Furthermore, models trained on limited data often lack the reasoning capabilities required for robust generalization to diverse real-world scenarios. To address these limitations, we leverage a powerful pretrained Vision-Language Model (VLM) to achieve a more sophisticated semantic understanding of the prompt and guide the composition of multiple subjects. We employ the Qwen2.5-VL model (Bai et al., 2025) for its superior multimodal reasoning capabilities. The VLM processes both the textual prompt $\mathbf{C}_{\text{txt}}$ and the set of reference images $\mathcal{I}$ to generate semantically enriched text tokens, which we denote as $\mathbf{f}_{\text{txt}}$: $\mathbf{f}_{\text{txt}} = \text{VLM}_{\text{enc}}(\mathbf{C}_{\text{txt}}, \mathcal{I}) \in \mathbb{R}^{l' \times c}$. These tokens capture not only the textual semantics but also the visual concepts of the subjects, enabling a more holistic understanding of the user's intent. These VLM-enhanced tokens then replace the embeddings from a standard text encoder. The complete set of conditioning tokens $\mathbf{C}_{\text{ctx}}$ is subsequently formed by concatenating these enhanced text tokens with the individual subject tokens: $\mathbf{C}_{\text{ctx}} = [\mathbf{f}_{\text{txt}}; \mathbf{f}_1; \ldots; \mathbf{f}_N] \in \mathbb{R}^{(l' + N \cdot hw) \times c}$. By integrating the VLM's advanced reasoning, ID-COMPOSER achieves superior semantic alignment and identity preservation, resulting in higher-quality videos that faithfully adhere to complex, multi-subject prompts.

### 3.3 Training with Online Reinforcement Learning

**GRPO** We use Flow-GRPO (Liu et al., 2025a) to perform online reinforcement learning (RL) to further improve the video generation quality. For each condition $q$, GRPO (Guo et al., 2025) samples a group of outputs $\{o_1, o_2, \ldots, o_G\}$ from the old policy $\pi_{\theta_{old}}$ and then optimizes the policy model by maximizing the following objective:

$$\mathcal{J}_{GRPO}(\theta) = \mathbb{E}_{q \sim P(Q), \{o_i\}_{i=1}^G \sim \pi_{\theta_{old}}(O|q)}$$

$$\frac{1}{G} \sum_{i=1}^G \frac{1}{|o_i|} \sum_{t=1}^{|o_i|} \left\{ \min \left[ r_t^i(\theta) \hat{A}_{i,t}, \text{clip}\left(r_t^i(\theta), 1-\epsilon, 1+\epsilon\right) \hat{A}_{i,t} \right] - \beta \mathbb{D}_{\text{KL}} \left[ \pi_\theta \,\|\, \pi_{\text{ref}} \right] \right\},$$

$$\text{where} \quad r_t^i(\theta) = \frac{\pi_\theta(o_{i,t} \mid q, o_{i,<t})}{\pi_{\theta_{old}}(o_{i,t} \mid q, o_{i,<t})}.$$

$$(3)$$

the relative quality $\hat{A}_{i,t}$ of the i-th response is computed as

$$\hat{A}_{i,t} = \frac{r_i - \text{Mean}(\{r_1, r_2, \ldots, r_n\})}{\text{Std}(\{r_1, r_2, \ldots, r_n\})},$$

Flow-GRPO extends GRPO to the rectified flow setting, while convert flow matching's determinsitic generation process to a stochastic one by injecting noise at each step of the flow integration. The update rule is defined as:

$$\mathbf{x}_{t+\Delta t} = \mathbf{x}_t + \left[ \boldsymbol{v}_{\boldsymbol{\theta}}(\mathbf{x}_t, t) + \frac{\sigma_t^2}{2t} \left( \mathbf{x}_t + (1-t) \boldsymbol{v}_{\boldsymbol{\theta}}(\mathbf{x}_t, t) \right) \right] \Delta t + \sigma_t \sqrt{\Delta t} \, \boldsymbol{\epsilon}, \qquad (4)$$

where $\boldsymbol{\epsilon} \sim \mathcal{N}(0, I)$ injects stochasticity into the generation process and $\sigma_t = a \sqrt{\frac{t}{1-t}}$.

**Reward** To further refine the model's output, we propose an online reinforcement learning (RL) phase subsequent to the initial RF training. This stage employs a bespoke policy-gradient algorithm to fine-tune the generative model $\mathbf{v}_{\theta}$. The objective is to maximize a composite reward function $\mathcal{R}_{\text{total}}$, defined as a weighted sum of video quality and identity consistency metrics:

$$\theta^* = \arg\max_{\theta} \mathbb{E}_{\mathbf{V} \sim p_{\theta}} \left[ \mathcal{R}_{\text{total}}(\mathbf{V}) \right], \tag{5}$$

where $\mathcal{R}_{\text{total}}(\mathbf{V}) := \lambda_{\text{qual}} \mathcal{R}_{\text{qual}}(\mathbf{V}) + \lambda_{\text{id}} \mathcal{R}_{\text{id}}(\mathbf{V}, \mathcal{I})$. Here, $\mathcal{R}_{\text{qual}}$ assesses the perceptual quality of the video, while $\mathcal{R}_{\text{id}}$ quantifies the identity preservation with respect to the reference images $\mathcal{I}$. The weights $\lambda_{\text{qual}}$ and $\lambda_{\text{id}}$ balance these two objectives.

## 3.4 TRAINING DATA CURATION

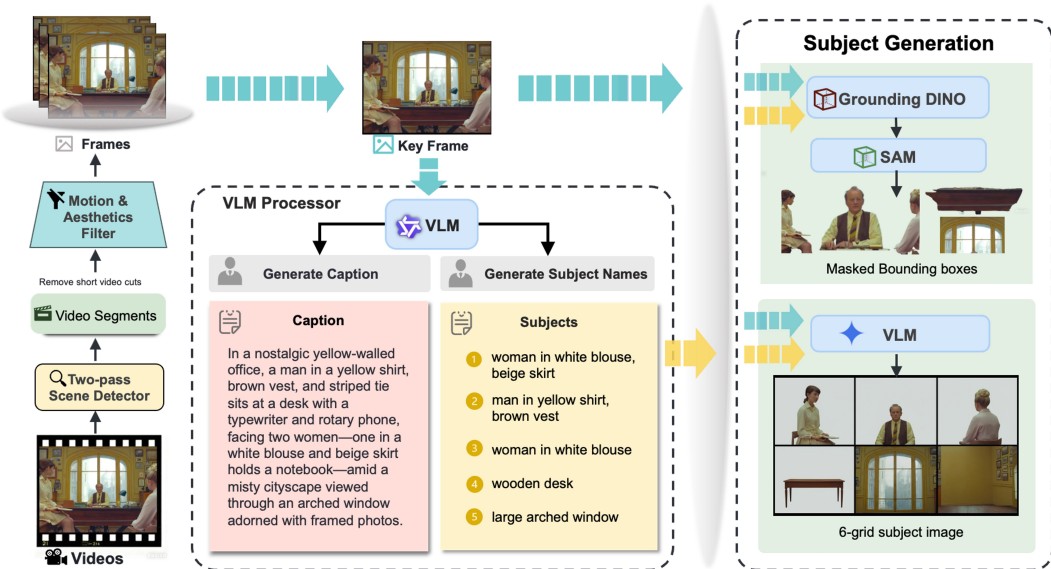

Figure 3: Data curation pipeline of ID-COMPOSER.

Current subject-to-video generation methods are hampered by notable deficiencies in output quality and diversity. These shortcomings are largely attributable to the restrictive nature of paired training data, which proves inadequate for modeling complex, real-world variations in subject motion, camera viewpoint, and scene layout. To surmount these limitations, we propose a sophisticated data-curation pipeline that harnesses the capabilities of modern Vision-Language Models (VLMs) (Bai et al., 2025) and powerful image editing models (Fortin et al., 2025). As shown in Fig. 3, the culmination of this pipeline is a large-scale, composite dataset meticulously constructed to propel the development of next-generation multi-subject video synthesis models. This dataset is composed of three heterogeneous data sources. The first component consists of subject-video pairs extracted from a diverse collection of OpenS2V-Nexus (Yuan et al., 2025), offering a wide array of authentic scenes and actions. The second component comprises synthetically generated data, where subjects rendered by cutting-edge image editing models are placed into novel contexts, systematically increasing the diversity of subject-background compositions. The third component is a high-fidelity collection of professionally shot videos with detailed annotations. More technical details are provided in Appendix C.

## 4 EXPERIMENTS

### 4.1 EXPERIMENTAL SETTINGS

**Implementation Details** Our model, ID-COMPOSER, is initialized from the weights of the Wan-Video-1.3 B model (Wan, 2025). To leverage superior semantic understanding, we employ a dual text-encoder architecture, combining T5 (Raffel et al., 2020) with Qwen2.5-VL-7B-Instruct (Bai et al., 2025). The model was trained for 30,000 iterations on our curated dataset at a resolution of 480p,

Table 1: **Quantitative Comparison against existing methods for the open-domain subject-to-video benchmark**. Total score is the normalized weighted sum of other scores. "↑" higher is better.

| Method | Total Score↑ | Aesthetics↑ | Motion↑ | FaceSim↑ | GmeScore↑ | NexusScore↑ | NaturalScore↑ |
|---|---|---|---|---|---|---|---|
| ***Proprietary Models*** | | | | | | | |
| Vidu 2.0 (Vidu, 2024) | 47.59% | 41.47% | 13.52% | 35.11% | 67.57% | 43.55% | 71.44% |
| Pika 2.1 (Pika, 2024) | 48.88% | 46.87% | 24.70% | 30.80% | 69.21% | 45.41% | 69.79% |
| Kling 1.6 (keling, 2024) | 54.46% | 44.60% | 41.60% | 40.10% | 66.20% | 45.92% | 79.06% |
| ***Open-source Models*** | | | | | | | |
| VACE-1.3B (Jiang et al., 2025) | 45.53% | 48.24% | 18.83% | 20.58% | 71.26% | 37.95% | 71.78% |
| VACE-14B (Jiang et al., 2025) | 52.87% | 47.21% | 15.02% | 55.09% | 67.27% | 44.20% | 72.78% |
| Phantom-1.3B (Liu et al., 2025b) | 50.71% | 46.67% | 14.29% | 48.55% | 69.43% | 42.44% | 70.26% |
| Phantom-14B (Liu et al., 2025b) | 52.32% | 46.39% | 33.42% | 51.48% | 70.65% | 37.43% | 68.66% |
| SkyReels-A2-P14B (Fei et al., 2025) | 49.61% | 39.40% | 25.60% | 45.95% | 64.54% | 43.77% | 67.22% |
| Ours-1.3B (Base Model) | 54.33% | 42.50% | 38.00% | 58.12% | 64.00% | 43.00% | 71.00% |
| Ours-14B | 57.05% | 45.00% | 40.00% | 60.50% | 67.00% | 45.00% | 73.00% |

utilizing 16 H20 GPUs. For inference, we employ an Euler sampler with 50 steps and classifier-free guidance (Ho & Salimans, 2022) to modulate the influence of image and text conditions, setting the classifier-free guidance scale to 2.5.

**Baselines** We benchmark our method against state-of-the-art open-source models that support native subject-to-video generation. This includes Phantom (Liu et al., 2025b) and SkyReels-A2 (Fei et al., 2025), both available in 1.3B and 14B parameter variants, and VACE (Jiang et al., 2025). For a comprehensive assessment, we also present a qualitative comparison with leading proprietary systems, such as VIDU (Vidu, 2024), Pika (Pika, 2024) and Kling (keling, 2024).

**Evaluation Metrics** Following the benchmark protocol of OpenS2V-Nexus (Yuan et al., 2025), we perform evaluation on a diverse dataset of **180 unique subject-text pairs**. We report on a suite of automated metrics, including Aesthetics, Motion quality, Face Similarity (FaceSim), and alignment scores against three benchmarks (GmeScore, NexusScore, NaturalScore). The Total Score is a normalized weighted sum of these metrics. Further details are provided in Appendix D.

## 4.2 MAIN RESULTS

**Quantitative Comparison** As demonstrated in Table 1, ID-COMPOSER achieves state-of-the-art performance across all evaluation metrics, outperforming existing open-source models by a significant margin. Notably, our method shows substantial improvements in FaceSim and NexusScore, which we attribute to our hierarchical identity-preserving attention mechanism and the rich semantic guidance from the VLM encoder. These components work in concert to ensure high-fidelity identity preservation and robust alignment with complex textual prompts.

**Qualitative Comparison** Figure 4 presents a qualitative comparison against leading methods. ID-COMPOSER consistently generates videos with superior visual quality, temporal coherence, and identity preservation. While baseline methods often exhibit artifacts or identity drift, especially in longer videos or complex scenes, our model maintains a stable representation of all subjects, accurately reflecting their appearance from the reference images.

## 4.3 ABLATION AND ANALYSIS

We conduct a series of ablation studies to validate the effectiveness of our key design choices. The results, summarized in Table 2, underscore the importance of each proposed component.

**Effect of Hierarchical ID-Preserving Attention** To isolate the contribution of our attention mechanism, we replaced it with a standard cross-attention mechanism that simply concatenates all subject and text tokens. As shown in Table 2, this variant ("*w/o* Hierarchical Attention") suffers a significant drop in FaceSim, confirming that our hierarchical approach is crucial for modeling subject-specific features and preventing identity degradation.

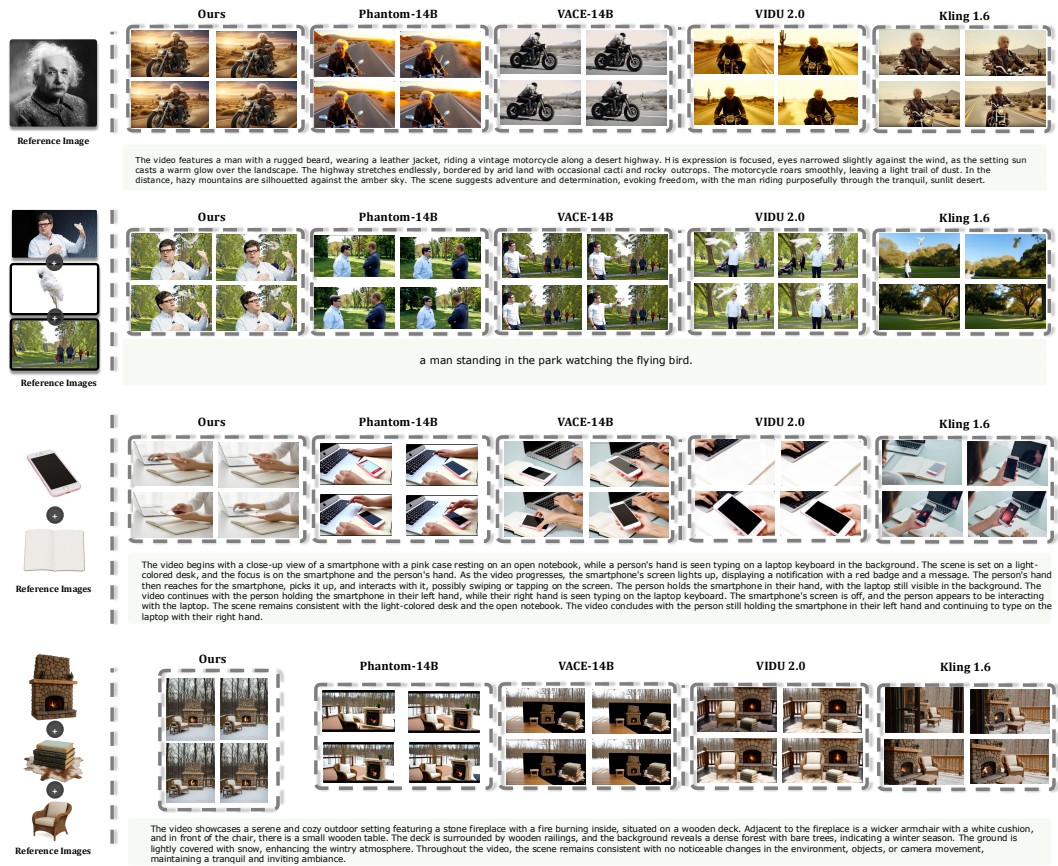

Figure 4: **Qualitative Comparison with State-of-the-Art Methods.** Our method, ID-COMPOSER, demonstrates superior performance in identity preservation, temporal consistency, and alignment with the textual prompt compared to existing open-source and proprietary models.

Table 2: **Ablation studies of ID-COMPOSER**. We evaluate the impact of our key contributions: the hierarchical identity-preserving attention, the VLM for semantic understanding, and our curated dataset. All metrics are reported on our evaluation set. "↑" indicates that higher is better.

| Method | FaceSim↑ | Text-Video Align.↑ | Video Quality↑ | Total Score↑ |
|---|---|---|---|---|
| ID-COMPOSER (Full Model) | **58.12%** | **49.55%** | **48.91%** | **54.33%** |
| w/o Hierarchical Attention | 51.34% | 48.98% | 47.52% | 50.11% |
| w/o VLM Encoder | 56.98% | 42.17% | 46.88% | 49.89% |
| w/o Curated Data | 54.55% | 45.32% | 45.13% | 48.78% |

**Effect of VLM-based Semantic Understanding** We evaluated the impact of the VLM by replacing the dual-encoder setup with a single T5 encoder (Raffel et al., 2020). The results ("w/o VLM Encoder") show a marked decrease in Text-Video Alignment. This highlights the VLM's superior ability to parse complex, multi-subject prompts and provide fine-grained semantic guidance, which is essential for accurate video generation.

**Effect of Data Curation Pipeline** Finally, we trained the full model on a baseline dataset without our curated data. This version ("w/o Curated Data") yielded lower scores across all metrics, particularly in Video Quality. This demonstrates that our data curation pipeline, which synthesizes diverse and high-quality training examples, is vital for achieving robust and generalizable performance. Qualitative results in Figure 5 further illustrate these findings.

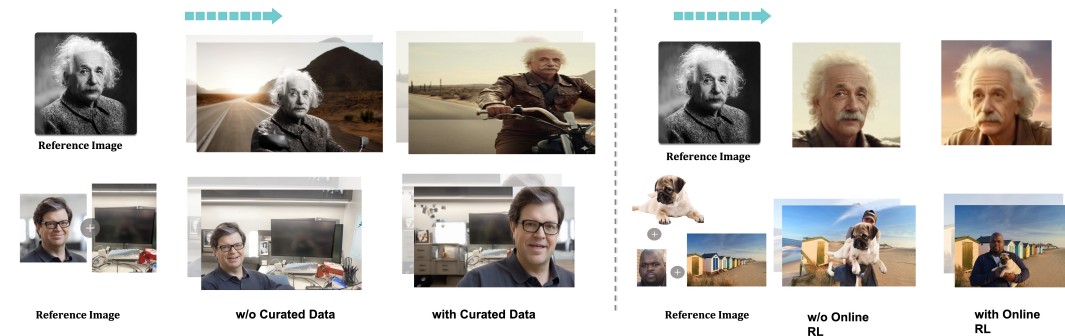

Figure 5: **Qualitative Ablation Study.** The left panel highlights the importance of our curated dataset, showing improved coherence and realism in subject integration compared to a model trained without it. The right panel illustrates the effectiveness of our online reinforcement learning stage, which significantly enhances visual quality and subject consistency.

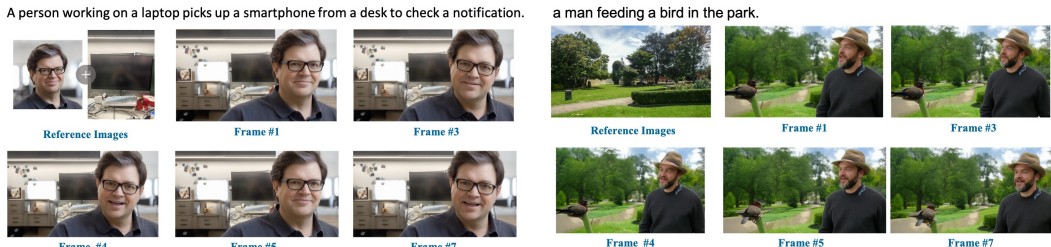

Figure 6: **Applications in Controllable Video Editing.** ID-COMPOSER enables zero-shot editing of existing videos, including subject replacement and background modification, while preserving identity and temporal consistency.

### 4.4 APPLICATION: CONTROLLABLE VIDEO EDITING

The architecture of ID-COMPOSER naturally lends itself to a range of editing applications. By providing an existing video as an initial state and conditioning the generation on new subjects or modified text prompts, our model can perform zero-shot, identity-consistent video editing. As illustrated in Figure 6, ID-COMPOSER can seamlessly insert new subjects into a scene, replace existing ones, or alter the background, all while maintaining temporal and semantic coherence. This capability opens up new avenues for personalized content creation and virtual storytelling.

## 5 CONCLUSION

In this work, we introduced ID-COMPOSER, an innovative framework for multi-subject video generation that integrates hierarchical identity-preserving attention and a pretrained Vision-Language Model into a latent diffusion backbone. Moving beyond simple feature concatenation, ID-COMPOSER captures complex intra- and inter-subject interactions, preserves identities, incorporates semantics, and ensures temporal consistency. Extensive experiments demonstrate that ID-COMPOSER achieves state-of-the-art performance, outperforming existing methods across a comprehensive suite of metrics. Our work marks an important step towards general and scalable multi-subject video generation in complex, real-world scenarios.

**Limitations and Future Work**  While ID-COMPOSER excels at identity preservation and multi-subject video generation, it remains limited by high computational costs and difficulties with complex interactions and fine-grained dynamics. Future work includes developing more efficient architectures, incorporating physics-aware priors and 3D reasoning, mitigating biases in pretrained components, and advancing controllable generation for fine-grained attribute, action, and interaction specification.

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

## A    DECLARATION OF LLM USAGE

During the writing of the manuscript, we utilized a Large Language Model (ChatGPT) as a writing assistant. The scope of its usage was limited to **improving grammar, polishing sentences, and enhancing the clarity and fluency of this manuscript**. The method, claims, experimental results and conclusions are developed by the authors.

## B    MORE IMPLEMENTATION DETAILS

**Reproducibility**    To facilitate reproducibility, we present our detailed experimental settings and evaluation metrics in Section 4.1. This section provides a comprehensive description of our implementation details. Moreover, **our source code and pre-trained models will be publicly available.**

### B.1    DETAILS OF NETWORK

We use a pre-trained VAE model from Wan-Video (Wan, 2025) to convert images from pixel space to latent space and vice versa. The latent representation has a down-sample ration of 8 and a latent channel of 16, and is then processed by a $2 \times 2$ patch embedding layer to reduce the spatial size and match the hidden dimension of the LLM backbone. The VAE model is frozen during training.

**Network Architecture**    Our video generation model is built upon three core modules: a Vision-Language Model (VLM) (Bai et al., 2025) for multi-modal comprehension, a Variational Autoencoder (VAE) for spatial compression, and a Multi-Modal Diffusion Transformer (DiT) (Wan, 2025) serving as the latent video diffusion backbone. The VLM, with a total of 7B parameters, is responsible for encoding textual prompts and subject images into a shared semantic space. It consists of a 32-layer Vision Transformer (ViT) and a 28-layer Large Language Model (LLM). The VAE is designed to encode video frames into a compact latent representation and subsequently decode them back to the pixel space. Its encoder and decoder utilize a spatial scale factor of $8 \times 8$, managing a parameter count of 54M and 73M, respectively. The core of our generative process is the DiT, a substantial 14B parameter transformer that operates on the latent codes produced by the VAE. With 60 layers and an intermediate size of 12,288, it is architected to model complex spatio-temporal dynamics for high-fidelity video synthesis. A detailed breakdown of the architectural configurations for each component is provided in Table 3.

Table 3: Detailed configuration of our model's primary architectural components. The VLM, VAE, and MMDiT modules are designed to handle multimodal understanding, spatial compression, and latent diffusion, respectively.

| Configuration | VLM | | VAE | | MMDiT |
|---|---|---|---|---|---|
| | ViT | LLM | Encoder | Decoder | |
| # Layers | 32 | 28 | 11 | 15 | 60 |
| # Num Heads (Q / KV) | 16 / 16 | 28 / 4 | - | - | 24 / 24 |
| Head Size | 80 | 128 | - | - | 128 |
| Intermediate Size | 3,456 | 18,944 | - | - | 12,288 |
| Patch / Scale Factor | 14 | - | $8\times8$ | $8 \times 8$ | 2 |
| Channel Size | - | - | 16 | 16 | - |
| **# Parameters** | 7B | | 54M | 73M | 1.3B / 14B |

### B.2 DETAILS OF REWARD CALCULATION

## C DATASET CURATION DETAILS

### C.1 DATASET STATISTICS

Figure 7 illustrates the composition and distribution of scenarios in our training dataset, whcih is meticulously curated from three primary sources to ensure comprehensive coverage and high fidelity. The foundational component is the large-scale OpenS2V dataset (Yuan et al., 2025), which contributes a substantial volume of 218,230 videos, accompanied by 535,259 masked subject images and 3,098 generated counterparts. To further elevate the dataset's quality, we incorporate a high-quality collection, comprising 9,374 pristine videos and an additional 3,000 generated subject images. Finally, to provide rich priors for appearance synthesis, we integrate a dedicated Subject Image dataset. This static collection contains 14,976 images, which are further categorized into 2,877 human-centric images and 3,458 isolated clothing items.

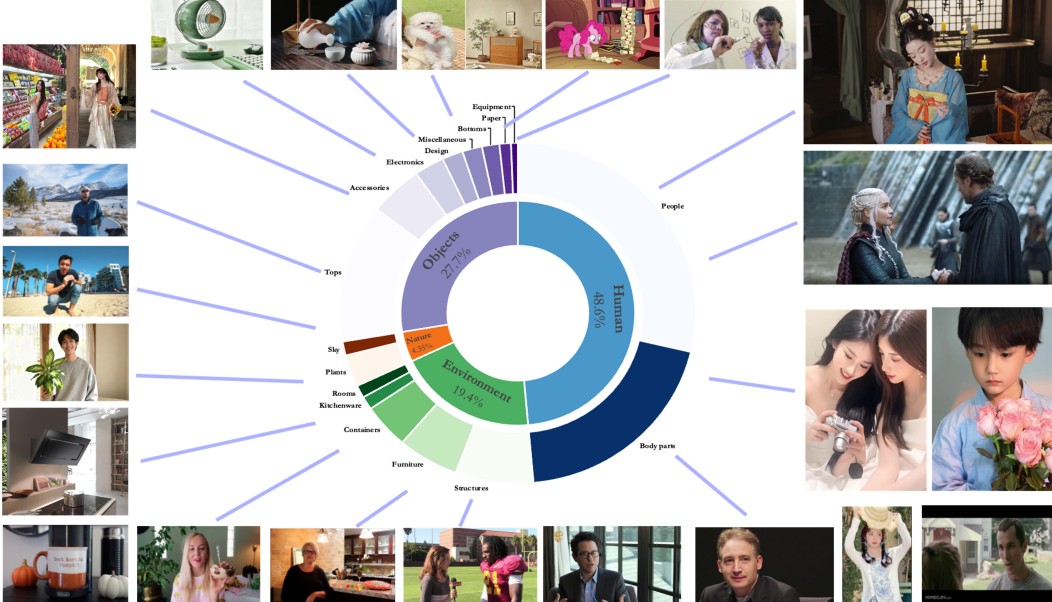

Figure 7: Statistics of the constructed dataset. The dataset is organized into four primary scenarios: Human, Objects, Environment, and Nature, each containing a variety of subcategories.

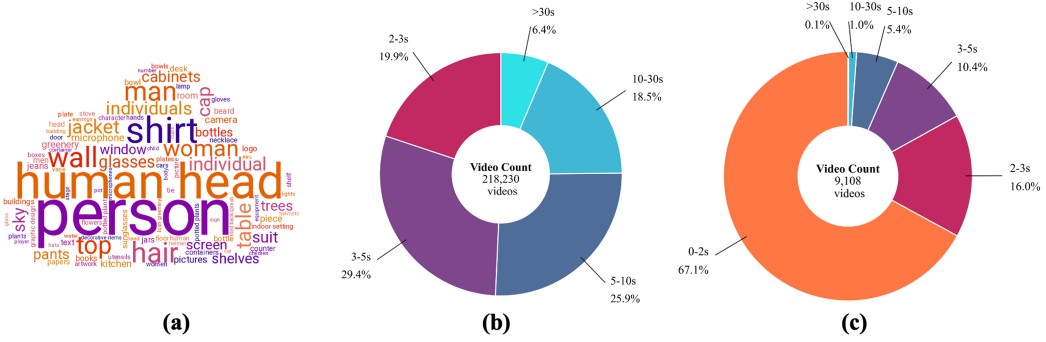

Figure 8: Statistics of the constructed dataset. (a) A word cloud illustrates the rich linguistic diversity of the reference images. The video duration distributions are shown for the two sources: (b) OpenS2V and (c) our curated videos.

### C.2 COMPARISON OF EXISTING DATASET

We analyze various strategies for constructing the reference dataset. As shown in Figure 9, Naïve data augmentation offers limited visual diversity and is prone to generating occluded subjects, which compromises generalization to complex scenes. Alternatively, employing on-the-shelf image-consistent generators, such as Flux-Kontext (Labs et al., 2025) or even GPT-4o (Hurst et al., 2024), often introduces undesirable appearance artifacts and **fails to maintain subject fidelity**.

In contrast, our approach, which leverages a tailored in-context learning prompt for the Gemini-Flash-Image model (Fortin et al., 2025), excels at subject decomposition and synthesis, yielding more faithful and varied reference images.

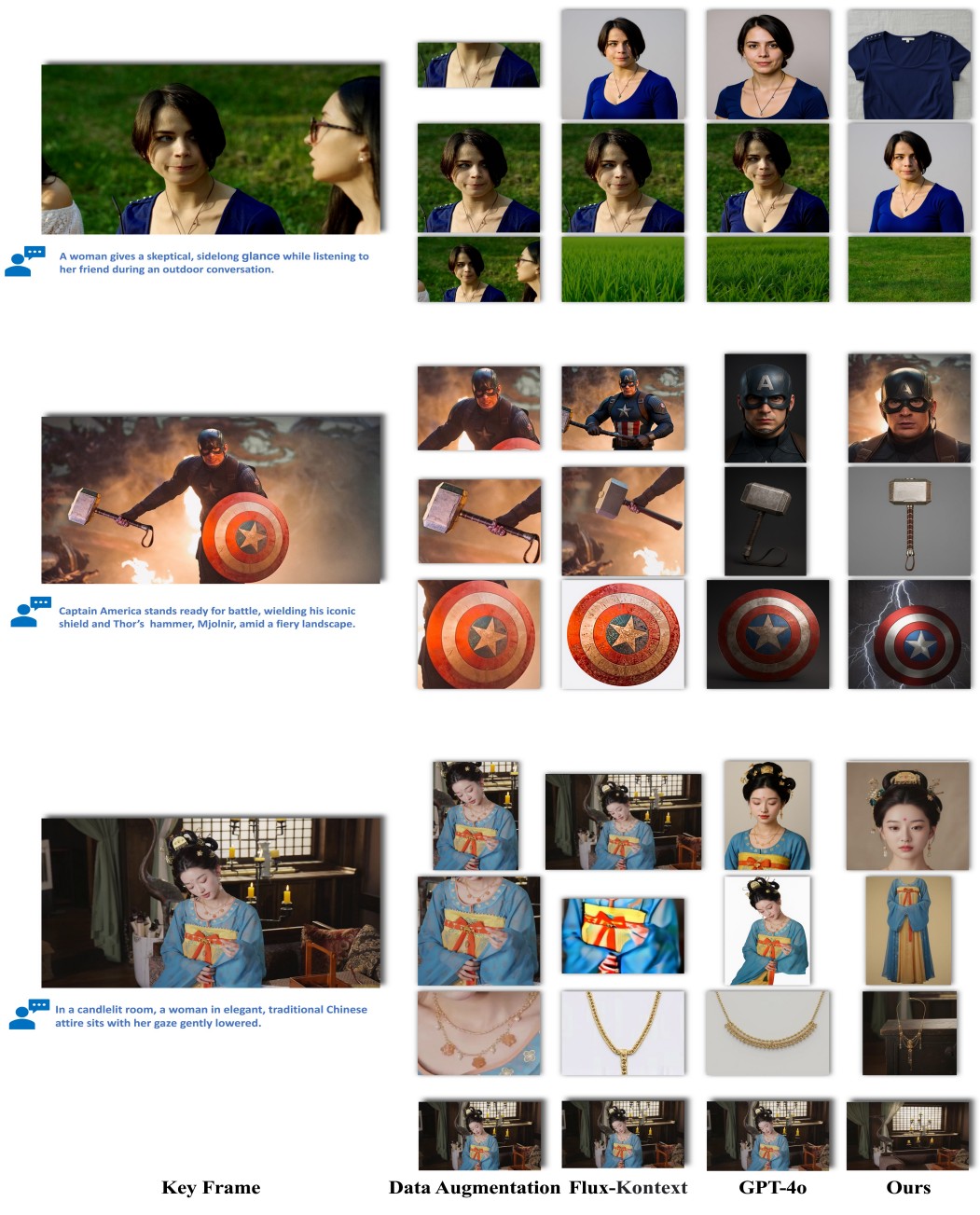

| Key Frame | Data Augmentation | Flux-Kontext | GPT-4o | Ours |

Figure 9: **Comparison of reference image construction strategies.**

# D    DETAILS OF EVALUATION METRICS

**NexusScore:** This metric quantifies the subject consistency between a generated video $\mathbf{V} = \{f_1, \dots, f_T\}$ and a reference image $\mathbf{I}_{\text{ref}}$. It employs a two-stage pipeline that first leverages Grounded-SAM to generate a subject mask $M_t$ for each frame $f_t$. Subsequently, a refined CLIP-based image encoder $\mathcal{E}_{img}$ is used to compute the cosine similarity between the feature embedding of the reference image and that of the masked subject in each frame. This targeted approach ensures the evaluation focuses strictly on identity fidelity by mitigating background interference. The final score is the average similarity over all $\mathbf{T}$ frames, formulated as: $S_{\text{Nexus}} = \frac{1}{\mathbf{T}} \sum_{t=1}^{\mathbf{T}} \frac{\mathcal{E}_{\text{img}}(\mathbf{I}_{\text{ref}}) \cdot \mathcal{E}_{\text{img}}(f_t \odot M_t)}{\|\mathcal{E}_{img}(\mathbf{I}_{\text{ref}})\| \|\mathcal{E}_{\text{img}}(f_t \odot M_t)\|}$ where $\odot$ denotes element-wise multiplication.

**NaturalScore:** This metric assesses the perceptual realism and physical plausibility of a generated video $\mathbf{V}$. We utilize a state-of-the-art Multimodal Large Language Model (MLLM) to perform a deep semantic analysis. The model is presented with the video and a carefully designed prompt which instructs it to evaluate spatio-temporal consistency, identify violations of physical laws, and detect visual artifacts characteristic of generative models. The model's direct output, a normalized score, serves as the final measure of the video's naturalness.

**GmeScore:** This metric evaluates the semantic alignment between a generated video $\mathbf{V}$ and its corresponding text prompt $\mathbf{C}$, proposed as an enhanced alternative to conventional CLIPScore. To overcome the limitations of prior methods in handling long-form text, GmeScore leverages the advanced text comprehension capabilities of the Qwen2-VL MLLM (Bai et al., 2025), which provides a powerful text encoder $\mathcal{E}_{text}$ and a temporally-aware video encoder $\mathcal{E}_{video}$. Unlike methods that average per-frame similarities, GmeScore computes a global feature representation for the entire video, thereby capturing its holistic temporal dynamics. The final score is the cosine similarity between the global video embedding and the text embedding: $S_{\text{Gme}} = \frac{\mathcal{E}_{\text{video}}(\{f_t\}_{t=1}^{T}) \cdot \mathcal{E}_{\text{text}}(C)}{\|\mathcal{E}_{\text{video}}(\{f_t\}_{t=1}^{T})\| \|\mathcal{E}_{\text{text}}(C)\|}$ This formulation provides a more accurate and comprehensive assessment of text-to-video relevance, especially for complex narratives.

# E    MORE COMPARISON RESULTS

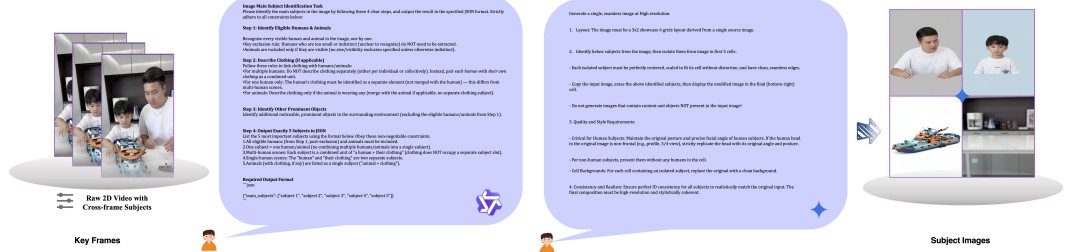

Figure 10: Prompt for MLLM-based multi-subject images generation.

We have provided video results on the in-the-wild videos in the supplementary material. Kindly refer to our website by opening the **index.html** file into a modern browser.

