# OpenReview forum: "ID-Composer: Multi-Subject Video Synthesis with Hierarchical Identity Preservation"
_ICLR.cc/2026/Conference — ICLR 2026 Conference Withdrawn Submission_

### Official Review · Reviewer_T9qL · 2025-10-20

**Soundness:** 2
**Presentation:** 3
**Contribution:** 2
**Rating:** 4
**Confidence:** 4

**Summary:**

This paper proposes ID-Composer, a framework for multi-subject video generation. The approach consists of (1) a hierarchical identity-preserving attention mechanism, (2) semantic understanding via pretrained vision-language model (VLM), (3) a data curation pipeline, and (4) an online reinforcement learning (RL). Experiments follow the OpenS2V-Nexus evaluation protocol. The base video generation model is Wan Video 1.3B, and the VLM is Qwen2.5-VL-7B-Instruct.

**Strengths:**

1. The overall method is clearly described and easy to follow.
2. The experiments include comparisons with both closed-source and open-source models.
3. A project website is provided for direct visual comparison of generated videos.

**Weaknesses:**

1. The authors have not reported the generation time complexity of the proposed method. Given that the hierarchical identity-preserving attention involves three operations, i.e., intra-subject attention, inter-subject attention, and multi-modal attention, it is likely to incur additional inference/generation time compared to the base model.
2. The method uses RL for optimization, but there is neither an explanation nor a comparison with SFT. In addition, while the right panel of Figure 5 appears to show results with and without RL, this is not discussed in the main text. Quantitative ablation results without RL are also missing from Table 2.

**Questions:**

1. The authors describe the training datasets in detail, but the evaluation dataset of 180 unique subject–text pairs is not well described. Could the training and evaluation datasets overlap? If not so, what have the authors done to avoid data leakage?

---

### Official Review · Reviewer_thth · 2025-10-25

**Soundness:** 3
**Presentation:** 4
**Contribution:** 3
**Rating:** 6
**Confidence:** 5

**Summary:**

The paper presents ID-Composer, a framework for multi-subject video generation based on text prompts and reference images. The method introduces a hierarchical identity-preserving attention mechanism to integrate subject features. Additionally, it leverages a pretrained vision-language model to enhance semantic understanding and capture complex multi-subject interactions. To better improve identity consistency, the authors further incorporate an online reinforcement learning stage. Experiments show that ID-Composer outperforms existing approaches in identity preservation, temporal consistency, and video quality.

**Strengths:**

1. By reading the related work section, I found that the authors are very familiar with the field of subject-driven video generation, which is the focus of this work.

2. The paper is clearly written and easy to follow. The authors provide sufficient background and motivation, and the structure of the paper is logical and coherent. The experimental design is sound, with comprehensive evaluations on relevant benchmarks, and the results convincingly demonstrate the effectiveness of the proposed approach. Overall, the experiments are thorough and well-presented.

**Weaknesses:**

1. Flow-GPRO evaluates the advantage function using only noise-free samples. However, this important detail is not introduced or discussed in Section 3.3.

**Questions:**

1. The paper claims to integrate the VLM’s advanced reasoning (e.g., line 242). However, to my understanding, ID-Composer only leverages the VLM to extract semantic features from text and images, without performing any explicit reasoning process (such as generating additional textual descriptions or conducting multi-step inference).

2.  The computation of the advantage function is confusing: the right-hand side seems to be defined over entire trajectories, yet the left-hand side, $\hat{A}^i_t$, is indexed by $t$. This discrepancy makes it unclear whether the advantage is intended to be computed per token or per trajectory. Although *DeepSeekMath*[1] adopts a similar formulation, they also define $\hat{A}^i_t = \tilde{r}_i$.

3. Is the hierarchical identity-preserving attention mechanism realized through 3D self-attention?

 If the authors can address my concerns, I would consider raising my score.

[1] DeepSeekMath: Pushing the Limits of Mathematical Reasoning in Open Language Models

---

### Official Review · Reviewer_UssJ · 2025-10-31

**Soundness:** 3
**Presentation:** 2
**Contribution:** 2
**Rating:** 4
**Confidence:** 4

**Summary:**

This paper addresses the problem of multi-subject video personalization. The proposed approach incorporates several components, including a hierarchical identity-preserving attention mechanism, VLM-based understanding, and an online reinforcement learning stage.

**Strengths:**

- The method is technically sound and leverages strong foundation models, with modules specifically designed to enhance identity coherence.
- The experimental results demonstrate satisfactory performance.

**Weaknesses:**

- The paper offers limited novelty, as all three modules—the attention mechanism, VLM encoder, and GPRO post-training—are relatively standard in the field.
- The use of VLM encoders for improved visual understanding is already common in current image and video foundation models.
- The GPRO section mostly presents established results with minimal innovation, primarily adding identity loss to an existing framework.
- Minor: lack related work reference [1].

[1] Wang, Zhao, et al. "Customvideo: Customizing text-to-video generation with multiple subjects." arXiv preprint arXiv:2401.09962 (2024).

**Questions:**

N/A

---

### Official Review · Reviewer_ZFyE · 2025-11-01

**Soundness:** 2
**Presentation:** 2
**Contribution:** 2
**Rating:** 2
**Confidence:** 5

**Summary:**

The paper proposes ID-COMPOSER, a multi-subject video generation framework that combines hierarchical identity-preserving attention, a pretrained vision-language encoder, and an online reinforcement learning (Flow-GRPO) phase to enhance semantic alignment and video quality. While the design is conceptually sound and well-written, several technical concerns weaken the contribution’s clarity and empirical depth.

**Strengths:**

1. Clear architecture combining hierarchical attention and VLM-guided semantics.
2. Comprehensive experiments with both open-source and proprietary baselines.
3. Demonstrated potential for controllable multi-subject editing.

**Weaknesses:**

1. The introduction of Flow-GRPO lacks a clear motivation beyond “improving video quality.” The paper does not specify the design of the reward components, their relative weights, or how they interact (e.g., identity vs. perceptual quality). Furthermore, there is no ablation study showing the effect of each reward or comparison with standard flow training.

2. The generated videos often exhibit visible copy-paste artifacts, particularly when multiple subjects are composited into dynamic scenes. This likely stems from the lack of cross-pair or mixed-subject training data, which limits the model’s ability to synthesize coherent multi-entity interactions. The data curation pipeline should explicitly include cross-subject compositions or synthetic fusion examples.

3. Several generated results (e.g., the “LeCun” case) show minimal motion and low temporal dynamics. The model tends to produce static or near-still outputs, suggesting that the training data or reward structure encourages conservative predictions.

**Questions:**

As seen in weakness

---

### Note · Authors · 2025-11-12

I have read and agree with the venue's withdrawal policy on behalf of myself and my co-authors.